# GRASP: Navigating Retrosynthetic Planning with Goal-driven Policy

**Yemin Yu[1,2], Ying Wei[1✉], Kun Kuang[2], Zhengxing Huang[2], Huaxiu Yao[3], Fei Wu[2,4✉]**

[1]City University of Hong Kong,  [2]Zhejiang University
[3]Stanford University,  [4]Shanghai Artificial Intelligence Laboratory
[1]{yeminyu2-c,yingwei}@cityu.edu.hk
[2]{kunkuang,zhengxinghuang,wufei}@zju.edu.cn
[3]{huaxiu}@cs.stanford.edu

## Abstract

Retrosynthetic planning occupies a crucial position in synthetic chemistry and, accordingly, drug discovery, which aims to find synthetic pathways of a target molecule through a sequential decision-making process on a set of feasible reactions. While the majority of recent works focus on the prediction of feasible reactions at each step, there have been limited attempts toward improving the sequential decision-making policy. Existing strategies rely on either the expensive and high-variance value estimation by online rollout, or a settled value estimation neural network pre-trained with simulated pathways of limited diversity and no negative feedback. Besides, how to return multiple candidate pathways that are not only diverse but also desirable for chemists (e.g., affordable building block materials) remains an open challenge. To this end, we propose a Goal-dRiven Actor-critic retroSynthetic Planning (GRASP) framework, where we identify the policy that performs goal-driven retrosynthesis navigation toward a user-demand objective. Our experiments on the benchmark Pistachio dataset and a chemists-designed dataset demonstrate that the framework outperforms existing state-of-the-art approaches by up to 32.2% on search efficiency and 5.6% on quality. Remarkably, our user studies show that GRASP successfully plans pathways that accomplish the goal prescribed with a goal (building block materials).

## 1 Introduction

Retrosynthetic planning has significantly advanced chemical synthesis, bringing in increasingly sophisticated medicines that cure diseases and materials that improve life. A retrosynthetic planner takes the structure of a target molecule as input and recursively selects feasible reactions to unsolved intermediate molecules until eventually reaching building block molecules. Since an unsolved intermediate molecule usually requires multiple steps of reactions to synthesize and at each step has up to hundreds of feasible reaction candidates, retrosynthetic planning with an enormous search space is very challenging even for experienced chemists. Consequently, computer-aided synthesis planning (CASP) enters the scene to assist chemists in accelerating the process of designing retrosynthetic pathways.

Computer-aided retrosynthesis planning consists of: 1) a single-step retrosynthesis prediction which predicts a list of feasible reaction candidates that connect a target molecule to its respective precursors, and 2) a multi-step planning policy that searches for the optimal synthetic pathway by recursively applying the single-step prediction model. Recent years have witnessed a plethora of advancements in single-step prediction models [25, 31, 19, 10], while in this work we are pursuing a more efficient and

36th Conference on Neural Information Processing Systems (NeurIPS 2022).

effective planning policy that limits the effective search space to include the most likely successful pathways.

Prior multi-step planning centered around tree or graph search methods [11, 21], where the search is guided by only the total reaction cost (quality) from the target molecule to the current node. To improve the search policy, recent attempts [24, 2, 7] include the estimated value from the current node to building block molecules, i.e., being building block aware. Unfortunately, the value estimation in [24, 7] is by online roll-out, unfavorably being of high variance and low search efficiency. Though [2] addressed this issue by pre-training a value network on simulated pathways, these pathways constructed from an existing single-step reaction dataset offer limited diversity and no negative experiences to learn from.

Over and above, scoring and ranking the quality of many feasible pathways towards a target molecule has been notoriously difficult. The considerations that dictate a high-quality pathway, including high reaction yields, simple reaction conditions, and low building block molecule costs, are oftentimes conflicting and require a trade-off; moreover, predicting reaction yields [22] and conditions is very challenging due to ill-defined and noisy annotations. Chen et al. [2] proposed to evaluate the quality with the negative log-likelihood of all reactions predicted by the single-step model, while it is predicated on the assumption that frequent reactions are with high yields or easy conditions and biased by the seen reactions that train the single-step model. Keeping in mind that the objective of retrosynthetic planning is to assist chemists, and in practice the challenge of quality evaluation can be overcome by 1) returning as diverse feasible pathways as possible for chemists to weigh their preferences, and 2) returning the pathways that meet the qualifying conditions prescribed by chemists, e.g., a set of very cheap building block materials or easy-to-synthesize intermediate molecules.

Therefore, we are motivated to propose a Goal-dRiven Actor-critic retroSynthetic Planning (GRASP) framework. Specifically, we formulate retrosynthesis planning as a reinforcement learning (RL) problem, where we first learn a policy network that takes continuous actions encoding the structure-level molecular information to allow navigation in the huge discrete action space of single-step reaction candidates. Moreover, GRASP learns a goal-driven $Q$-value estimation network to update the policy, by sampling both successful (positive) and failed (negative) experiences and relabeling the goals of sampled experiences. Finally, the learned Q-value estimation and policy networks join to guide the Monte-Carlo Tree Search, after which GRASP returns diverse pathways as a result of a good exploration-exploitation tradeoff. In summary, our contributions are threefold.

- We propose a novel actor-critic retrosynthetic planning framework GRASP, which learns from extensive positive and negative experiences to navigate through huge single-step reaction spaces.
- We are the first to empower goal-driven planning, which mitigates the challenge in quality evaluation of pathways by directly fulfilling the requirements prescribed by chemists.
- We have evaluated the performance of GRASP on both an academic and an industrial benchmark dataset. The results and user studies demonstrate that GRASP outperforms all baselines in general retrosynthetic planning metrics by a significant margin and is the first to achieve high-quality goal-driven retrosynthetic planning.

## 2 Related Work

**Single-step Retrosynthesis Prediction** Single-step prediction models can be categorized into two main classes, i.e., template-based and template-free. Template-based methods rely on templates that encode chemical reaction cores to convert a product molecule into reactants. The key is to rank templates and select an appropriate template to apply, for which recent attempts [3, 23] solve the problem of template selection through a classification neural network. Despite their superior interpretability, template-based approaches are disadvantaged by 1) the daunting challenge of atom-mapping for template extraction, and 2) poor generalization to unknown reaction types or structures beyond templates. On the other hand, template-free methods [14, 13, 20], inspired by the recent progress of *seq2seq* [27] and *Transformer* [28], regard single-step retrosynthesis prediction as a translation task and translate a product molecule represented in SMILES strings [29] to reactant SMILES strings. To join the benefits of template-based and template-free methods, recent works [25, 31, 19] seek semi-template-based methods where the reaction center dictating a reaction is firstly predicted via graph neural networks and the resulting intermediate synthons are secondly translated into reactants via seq2seq or graph translation models. Recently, Kim et al. [10] proposed to

fine-tune a single-step prediction model with the feedback from a multi-step retrosynthetic planning policy, leading to a search-guided single-step model. We have validated in Section 4.3 that our planning policy is also compatible with the framework and improves single-step prediction and thereby final pathways.

| | Neural-guided search | building blocks awareness | Negative experiences | Exploration-exploitation tradeoff | Goal-driven |
|---|---|---|---|---|---|
| HgSearch [21] | ✗ | ✓ | ✗ | ✓ | ✗ |
| DFPN-E [11] | ✗ | ✗ | ✗ | ✗ | ✗ |
| MCTS [24] | ✗ | ✓ | ✓ | ✓ | ✗ |
| Retro* [2] | ✓ | ✓ | ✗ | ✗ | ✗ |
| Ours | ✓ | ✓ | ✓ | ✓ | ✓ |

Table 1: Comparison of different planning frameworks in five dimensions. **Neural-guided Search** learns from past multi-step planning experiences a planning policy characterized by a neural network; **Building blocks Awareness**: The value of a planning policy is biased towards reactions leading to building block molecules; **Negative Experiences** mean planning pathways with failure; **Exploration-exploitation tradeoff**: A planning policy balances exploration and exploitation, resulting in more diverse pathways. **Goal-driven**: A planning policy is capable of performing planning towards a specific goal.

**Multi-step Retrosynthetic planning** We summarize the comparison of existing multi-step retrosynthetic planning policies in Table 1. Specifically, previous planning methods HgSearch [21] and the proof number search [11] are traditional heuristic search algorithms, where the chemical feasibility and the value of the negative (failure) pathways are not considered. Inspired by AlphaGo [26], Segler et al. [24] adopted the Monte-Carlo tree search to generate a search tree on the fly and explore and generate multiple synthesis pathways. Unfortunately, each node with a combination of all precursors in a reaction leads to enormous search space, and the value estimation by vanilla online roll-out is of high variance and high computation cost. Inspired by [11], Chen et al. [2] designed a neural-based A*-like algorithm that learns an additional value network with **automatically constructed and only successful** pathways to bias the search prior. Recently, Han et al. [9] and Xie et al. [30] used a GNN-based value network to capture inter-molecular/intra-pathway level information to further improve the A*-like retrosynthetic planning algorithm. However, one disadvantage of A*-like [2] retrosynthetic planning algorithm is that it fails to balance exploration and exploitation, resulting in less diverse pathways. Moreover, none of the previous approaches is capable of biasing the retrosynthetic planning toward a favorable goal prescribed by chemists.

## 3 Methods

First, Sec. 3.1 introduces Markov decision process (MDP) setting for goal-driven retrosynthetic planning. Secondly, Sec. 3.2 elaborates on the framework of the GRASP goal-driven actor-critic agent and the training procedure with TD3 [6] algorithm. Lastly, Sec. 3.3 introduces GRASP planning for a given target molecule under a goal-driven variant of MCTS.

### 3.1 Goal-driven MDP for retrosynthetic planning

We denote a finite-horizon MDP by $\mathcal{M} = \{\mathcal{S}, \mathcal{A}, \mathcal{T}, \mathcal{G}, r(s, a, g), H, \gamma\}$ for our goal-driven retrosynthetic planning task. We use $s \in \mathcal{S}$ to denote the state (molecule) space, $a \in \mathcal{A}$ to denote the action (reaction) space which consists of reaction candidates $a$ generated by the single-step prediction model, and $\mathcal{T}(s_{t+1}|s_t, a_t)$ to denote the state transition from $s_t$ to $s_{t+1}$ through performing reaction $a$ with a deterministic state transition probability. We denote the goal space as $\mathcal{G}$, which has the same size as the state space $\mathcal{S}$ since our goal is to navigate toward particular states. Considering the ultimate goal for retrosynthesis is to discover retrosynthetic pathways reaching building block molecules, we denote the goal for the entire set of building block molecules as $\mathcal{G}_\mathcal{B}$, where each goal $g_i \in \mathcal{G}_\mathcal{B}$ indicates a specific building block molecule $i$. To simultaneously adapt to both general (non-goal-driven) and goal-driven retrosynthetic planning, we define $g = \mathcal{G}_\mathcal{B}$ as all zero embedding and concatenate the goal embedding with an additional binary feature embedding, where we use $\mathbf{I}(g = \mathcal{G}_\mathcal{B}) = 0$ for the general planning and $\mathbf{I}(g = g_i) = 1$ for the goal-driven planning towards goal $g_i$. For the reward design of $r(s, a, g)$, we assign the goal-driven path-finding reward as $r(s, a, g) = 1$ when the state $s$

reaches the desired goal $g$ after taking action $a$ and $r(s, a, g) = 0$ otherwise. Finally, $\gamma$ is the discount factor, and $H$ is the maximum horizon (length) for the pathway.

## 3.2 GRASP framework and training procedure

GRASP has two parameterized components as shown in Fig. 1: actor network denote by $\pi_\phi(a|s, g)$ and critic network by $Q_\theta(s, a, g)$. In the setting of retrosynthetic planning, we regard the upstream single-step retrosynthesis **predictor** as the **environment** and retrosynthetic **planner** as the **agent**. At each time step $t$, the agent outputs a proto-action $\tilde{a}_t$ with the same size as action embedding, according to its goal-driven policy network $\pi_\phi(\tilde{a}_t|s_t, g)$ from observing the current state $s_t$ and goal $g$. Since we are unaware of the possible goal state for a given initial state without prior knowledge, we use $g = \mathcal{G}_\mathcal{B}$ in $\pi_\phi(\tilde{a}_t|s_t, g = \mathcal{G}_\mathcal{B})$ as behavioral policy. Specifically, we add a small amount of random noises $\mathcal{N}$ to the action for exploration during sampling:

$$\tilde{a}_t{'} = \tilde{a}_t + \epsilon; \quad \epsilon \sim \mathcal{N}(\mu, \sigma).$$

After acquiring the proto-action $\tilde{a}_t{'}$, the agent has to identify an actual reaction $a_t$ from available reaction candidates $\mathcal{A}(s_t)$ given the state $s_t$ and action embedding $\tilde{a}_t$. Inspired by the k-nearest neighbor (k-NN) trick for large discrete action space similar to the Wolpertinger training [5], we use the true action embeddings from the available actions $\mathcal{A}(s_t)$ for the k-NN calculation during the action selection procedure. Furthermore, we may encounter reactions that induce more than one non-building block molecule as reactants, namely convergent synthesis. Convergent synthesis reaction, although infrequent in retrosynthesis , introduces a variation in the cardinality of state representation $s$ that conflicts with MDP settings. In previous work in MCTS for retrosynthesis, Segler et al. [24] accumulates all non-building block reactants as a set of molecules in state representation, but only performs action selection on a single molecule. The combinatorial nature of state representation introduces bias in reward propagation and sparsity in variance estimation. To overcome this complexity during the sampling and training phase, we use the average distance among all reactants for k-NN computation each time we encounter a reaction with convergent synthesis. As a result, we obtain an actual reaction $a_t$ by referring to the k-NN computation of the proto-action $\tilde{a}_t$ over the available actions $\mathcal{A}(s_t)$. If a convergent synthesis reaction is identified as the true action $a_t$ by the environment, non-building block reactants are split into separate next states as independent trajectories to perform parallel sampling. Eventually, the next state $s_{t+1}$ is defined as the non-building block molecule among the reactants of $a_t$. The sampling of a trajectory terminates when the state reaches the goal $g$ or the length of the trajectory reaches the maximum horizon $H$.

**Goal-driven relabeling:** To capture the goal-driven planning insights from a retrosynthesis pathway and accelerate learning in the sparse reward setting, we are inspired by [1] to relabel transition tuples in trajectories. The core idea of applying goal-driven relabeling in retrosynthesis is to exploit the data generated from the general retrosynthesis policy $\pi_\phi(a|s, g = \mathcal{G}_\mathcal{B})$ to train featured retrosynthesis planning data, and incorporate the agent with knowledge of navigating toward a specific goal state $g = g_i$. In practice, we copy the state transition tuple $\mathcal{M}_i = (s_i, a_i, r_i(.|g_i = \mathcal{G}_\mathcal{B}), s_{i+1})$ and randomly relabel the tuple $\mathcal{M}_i = (s_i, a_i, r_i(.|g_i'), s_{i+1})$ with a relabeling probability $p_r$ using *future* relabeling strategy. Specifically, for the $i$th tuple $\mathcal{M}_\rangle$ in trajectory $\tau$ with length $T$, we perform goal-driven relabeling by iterating over all future transitions as:

$$g_i' = \begin{cases} s_{i+k}, & p_r \\ \mathcal{G}_\mathcal{B}, & 1 - p_r \end{cases}$$

for $k \in (0, T - i]$. Since the relabeling probability $p_r$ is an important hyperparameter to balance between general and goal-driven planning, we will further examine the effect of different $p_r$ on the planning performance in the experiment.

The RL agent is trained with TD3 [6] algorithm. For tuple $i$ in a training batch, the target critic network is first updated using the one-step TD equation as:

$$y_i^{td} = r_i + \gamma Q'(s_{i+1}, \pi'(s_{i+1}, g_{i+1}), g_{i+1}), \tag{1}$$

where, $Q'$ and $\pi'$ denote the target critic and actor networks with fixed parameters copied from original critic and actor networks $Q_\theta$ and $\pi_\phi$ respectively, and $r_i, s_{i+1}, g_{i+1}$ represents the reward, state, and goal at the step $t$. With the TD target $y_i$, we can calculate the batch mean-square-error loss on the original critic network $Q_\theta(s, a, g)$ as:

$$L(\theta) = \frac{1}{N} \sum_i (y_i^{td} - Q_\theta(s_i, a_i, g)). \tag{2}$$

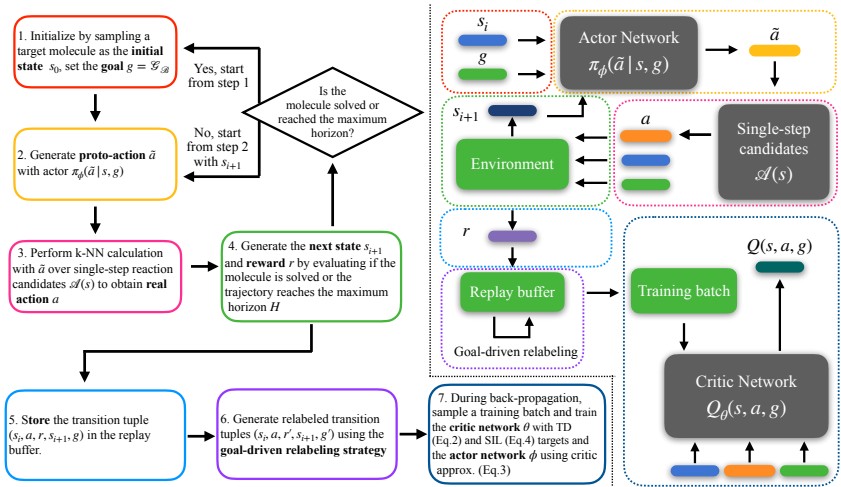

Figure 1: GRASP training flowchart (left) and the goal-driven actor-critic framework (right). Grey boxes indicate the agent-level components that will be used further during GRASP planning, and green boxes indicate the components used in GRASP training only.

Since the goal of the actor network is designed to maximize the overall return (success rate), and the goal of critic network is to approximate the overall return, the actor $\pi_\phi$ can be trained by maximizing the $Q$ value by minimizing:

$$L(\phi) = -\frac{1}{N} \sum_i (-Q_\theta(s_i, \pi_\phi(\tilde{a}|s_i, g), g)). \tag{3}$$

**Self-imitation learning:** To learn from highly imbalanced pathways in the overall search space ($> 85\%$ failures), we adopt self-imitation learning [17] (SIL) to accelerate the convergence in sparse reward and enhance the training efficiency. Intuitively, SIL assists the RL agent to emphasize high-quality planning experiences. Instead of using the Bellman equation for calculating the target Q-value, SIL directly uses the Monte-Carlo return of each 'good' episode as the Q-value target. It is crucial for the agent to exploit success trajectories in retrosynthetic planning tasks, especially during the early stage of training when a large proportion of samples in the replay buffer originated from failed trajectories. We denote the simplified SIL loss for $i$th tuple in a success trajectory $\tau$ with length $h$ as:

$$L(\theta) = \frac{1}{N} \sum_i (y_i^{sil} - Q_\theta(s_i, a_i, g)), \tag{4}$$

where $y_i^{sil} = \sum_{i=k}^h \gamma^{h-k} r_i$. We also include the full training algorithm in Alg. 1

### 3.3 GRASP retrosynthetic planning

In this section, we demonstrate the GRASP planning procedure for a target molecule and a specific goal with GRASP RL agent $\pi_\phi(a|s, g)$ and $Q_\theta(a, s, g)$.

Since each newly expanded molecule node is the same as the initial state in GRASP , it is natural to combine our RL agent into Monte-Carlo tree search (MCTS) with goal-driven p-UCT function [26]:

$$a_t = \underset{a \in \mathcal{A}(s_t)}{\operatorname{argmax}} \frac{Q(s_t, a, g)}{N(s_t, a)} + cP(a|s_t, g)\frac{\sqrt{N(s_{t-1}, a_{t-1})}}{1 + N(s_t, a)}. \tag{5}$$

In previous MCTS for retrosynthesis, Segler et al. [24] used an online roll-out stage for Monte-Carlo estimation of success rate for each leaf node, which both suffer from high-variance and heavy computation. Therefore, one of the key differences between GRASP and Segler et al. during the planning stage is we completely skip the online roll-out stage and directly refer to the RL agent for value estimation of the leaf nodes instead of Monte-Carlo estimation from the online roll-out. To align with our MDP settings in RL, we adopt a goal-driven MCTS planning with individual molecules

**Algorithm 1** GRASP

---

**Initialize** Critic network $Q_\theta, Q'_\theta$ and actor network $\pi_\phi, \pi'_\phi$, replay buffer $B$, initial state space $\mathcal{S}$, action space $\mathcal{A}$, goal space $\mathcal{G}$, reward function $r : \mathcal{S} \times \mathcal{A} \times \mathcal{G} \to \mathbb{R}$.
**for** $ep = 1$ **to** $M$ **do**
    Sample initial state $s_0 \in \mathcal{S}$.
    **for** $t = 0$ **to** $H$ **do**
        Sample proto-action $\tilde{a}_t$ using behavioral policy and exploration noise for general planning $\pi'_\phi(a|s_t, G_B)$
        Perform k-NN computation and execute action $a_t$
        Observe reward $r_t = r(s_t, a_t, G_B)$ and next state $s_{t+1}$
    **end for**
    **for** $t = 0$ **to** $H$ **do**
        Store original transition $(s_t, a_t, r_t, s_{t+1}, g)$ in $B$
        Generate transition copy and relabel $g' = s_{t+i}$ and $r'_t = r(s_t, a_t, g')$ with probability $p_r$ with *future* goal-driven relabeling strategy.
        Store transition $(s_t, a_t, r'_t, s_{t+1}, g')$ in $B$
    **end for**
    **for** $t = 0$ **to** $N$ **do**
        Perform actor-critic batched TD training on $\theta, \phi$ with Eq.2 and Eq.3
        Perform SIL training on $\theta$ with Eq.4
    **end for**
**end for**

---

as tree node representation. Specifically, our framework consists of three phases as shown in Fig. 2, and for simplicity, we ignore all building block molecules in the figure since no selection action will be performed on:

- **Selection:** Starting from the root node, the p-UCT function in Eq. 5 is used to iteratively select an action. At any step $t$, available actions in candidate set $a_t \in \mathcal{A}(s_t)$ and respective single-step confidence score $p_c(a_t|s_t)$ are provided by the single-step retrosynthesis predictor, and we define:

$$p(a_t|s_t, g) = p_c(a_t|s_t) \frac{exp(\frac{1}{D(\tilde{a}, a_t)})}{\sum_{a_j \in \mathcal{A}(s_t)} exp(\frac{1}{D(\tilde{a}, a_j)})}, \qquad (6)$$

where $D(\cdot)$ is the same distance metric used in the k-NN calculation, $\tilde{a}$ is produced by policy network $\pi_\theta(\tilde{a}|s_t, g)$, and $N(s_{t-1}, a_{t-1})$ denotes the visit count of the state-action pair of previous states. If a convergent synthesis action with multiple non-building block reactants is selected, we perform parallel selection and select all non-building block reactants as the next state. We iteratively perform selection on states until reaching a leaf node. Eventually, a set of leaf nodes is identified for expansion.

- **Expansion:** Each leaf node $s_t$ from the selected set is expanded by referring to the single-step model. Each available action from $\mathcal{A}(s_t)$ is directly appended to the node $s_t$. For convergent synthesis action, we generate the same number of leaf nodes depending on the quantity of unsolved molecules. For each newly generated leaf node, we evaluate their $Q^*$ value with the following rule: if there is no available action for $s_t$, we directly apply $Q^* = 0$. Before applying $Q_\theta(s, a, g)$ network to assign $Q^*$ value for newly added leaf nodes, we assign $Q^* = 1$ and label it as 'solved' if the $s_t$ reaches $g$ or $Q^* = 0$ if the state reaches the maximum horizon. If the state is undetermined, we assign $Q^* = Q_\theta(s_{t+1}, a, g)$ by applying $Q_\theta$ value network.

- **Update:** During the update phase, the $Q^*$ values and visit counts $N(s, a)$ are traversed backward following the selection path from leaf nodes back to the root node. We use a simple moving average for updating $Q$ value with a discount factor $\gamma$:

$$Q'(s_t, a, g) = Q(s_t, a, g) + \frac{1}{N(s, a)}[\gamma Q^* - Q(s_t, a, g)].$$

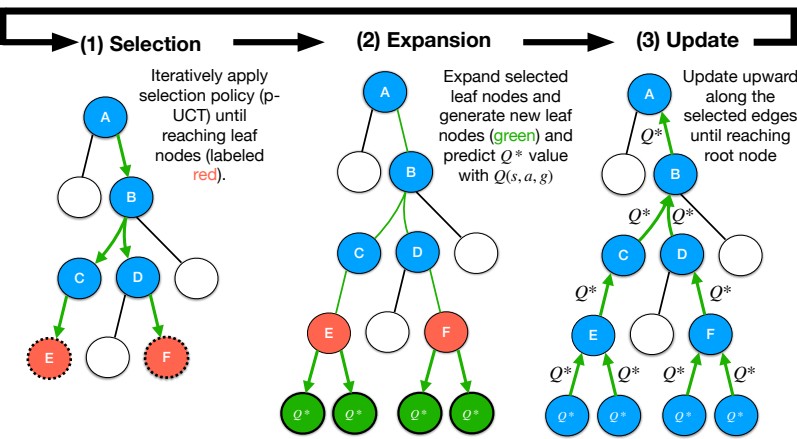

Figure 2: Overview of GRASP planning procedure. Specifically, the selected pathway (green) includes 4 specific reactions:$A \rightarrow B, B \rightarrow C + D, C \rightarrow E, D \rightarrow F$.

## 4 Experiments

### 4.1 Experiment setup

**Baseline Algorithms:** We compare our approach against a varieties of baselines including: 1. **MCTS[24]:** The original vanilla version of MCTS for retrosynthesis planning which exploits online roll-out to obtain Monte-Carlo estimation of future value without data generalization. 2. **DFPN-E[11]:** Depth-first proof number search (PNS) performed on AND-OR tree with an additive reaction likelihood as cost. 3. **Retro\* and Retro\*-0 [2]:** Different from DPFN-E, Retro\* utilizes the AND-OR tree as a single-player game and utilizes global value estimation. Additionally, Retro\* pre-trains a value network on a simulated retrosynthesis pathway dataset. Retro\*-0 denotes its version that performs the search without the value network. Retro\* is reported as the state-of-the-art search algorithm for retrosynthetic planning. 4. **Hyper-Graph Search (HgSearch)[21]:** HgSearch is a beam-search-like algorithm performed on a hyper-graph structure. The heuristics are the product of the single-step confidence score and molecular complexity score (SCScore) [4].

**Evaluation metrics:** We use four different metrics to comprehensively evaluate the performance of different search algorithms: 1. **Pathway length:** We use the total number of reactions in the retrosynthesis pathway for length evaluation. 2. **Pathway cost:** The cost function is defined as the summation of the negative log-likelihood (confidence score) of the reactions in the pathway $\tau$ provided by the single-step model, i.e., $-\sum_{a \in \tau} \log p_c(a|s)$ [2]. The cost is also regarded as a criterion for chemical feasibility. 3. **Planning efficiency:** Since the primary objective of AI-aided retrosynthesis is to help chemists find successful pathways faster, efficiency has been a crucial evaluation criterion for a multi-step retrosynthesis planning algorithm. Therefore, we follow [2] to take the *number of single-step inference calls* as a qualified surrogate of *time*, as single-step inference ($\sim$ 2s per iter) takes up almost $> 99\%$ of the time (only $\sim 0.006$s per iter on planning). 4. **Success rate:** With a fixed number of single-step inference calls, the success rate is defined as the percentage of solved molecules in the entire set.

**Single-step retrosynthesis predictor:** We adopt the template-free single-step retrosynthesis predictor based on molecular transformer (MT)[20, 13] from Schwaller et al. [21] as our single-step retrosynthesis predictor. Specifically, Schwaller et al. separately trained a pair of backward single-step generation models and forward single-step prediction models, and cooperatively utilized them to generate high-quality single-step retrosynthetic candidates with a confidence score $p_c(a|s)$ ranging from 0 to 1. Both statistics [14, 13] and our user study in real-world scenarios demonstrate that MT-based single-step framework achieves higher accuracy and less chemo-selectivity when compared with template-based approaches. Eventually, we choose top-$k$=100 reactions ranked according to the confidence score predicted by the single-step model as the available single-step candidate set for a given molecule since top-$k$=100 is sufficient to represent feasible single-step reaction space for a molecule.

| PISTACHIO | GRASP | RETRO* | RETRO*-0 | HGSEARCH | DFPN-E | MCTS |
|---|---|---|---|---|---|---|
| AVG. LENGTH | **4.12** | 4.27 | 4.25 | 4.38 | 4.22 | 4.74 |
| AVG. COST | 7.47 | 7.53 | 8.44 | **7.06** | 12.88 | 13.72 |
| AVG. TIME | **42.6** | 62.0 | 82.4 | 94.5 | 84.3 | 116.5 |
| SUCCESS RATE | **0.95** | 0.92 | 0.92 | 0.87 | 0.85 | 0.81 |

| WUXITEST | GRASP | RETRO* | RETRO*-0 | HGSEARCH | DFPN-E | MCTS |
|---|---|---|---|---|---|---|
| AVG. LENGTH | **6.93** | 7.50 | 7.38 | 7.65 | 7.29 | 8.15 |
| AVG. COST | **21.19** | 22.55 | 26.55 | 21.41 | 30.89 | 38.62 |
| AVG. TIME | **79.3** | 112.5 | 157.2 | 194.7 | 183.9 | 224.5 |
| SUCCESS RATE | **0.86** | 0.80 | 0.78 | 0.82 | 0.64 | 0.52 |

Table 2: General planning performance summary on Pistachio and WuxiTest. Average statistics is calculated among all successful pathways with $N_{max} = 400$ for both datasets.

## 4.2 Creating benchmark datasets

**Single-step reactions and building block molecules dataset:** We use the Pistachio reaction dataset (Ver. 18.11.19) [16] as our benchmark dataset for training our single-step models, and the implementation details are listed in Appendix A.1. After further pruning and discarding reactions with multiple products, the entire dataset consists of 2.7M reactions. The dataset is further split randomly into train/val/test sets following 90%/5%/5% proportions. We use the complete 231M commercially available molecules presented in *eMolecules* [1] for the building block molecule set.

**Pathway dataset:** Since only Retro* requires an additional simulated retrosynthesis pathway dataset for pre-training its value network for planning, we follow the setting in [2] and construct the Pistachio pathway dataset similarly. Specifically, we obtained 61554 pathways with an average length of 3.66. We split the dataset into 40000 training pathways, 21354 validation pathways, and 200 test pathways for Retro* value network training. Note that constructing an artificial pathway dataset by simply concatenating single-step reactions is only a reference rather than an optimal/expert pathway for a given molecule in a given search space. Moreover, an expert pathway dataset is unavailable for unreported molecules and expensive to obtain in real-world scenarios. Therefore for a fair comparison, the target molecules in the pathway dataset are simultaneously used as initial states for GRASP training.

**Expert dataset:** We also include a real-world expert dataset 'WuxiTest' designed by WuxiAppTec chemists, and each target molecule is provided with one reference pathway. WuxiTest consists of 500 molecules that are specifically designed to consist only of molecules that have never appeared in any journals and patents. Molecules were split into ten categories in terms of retrosynthesis strategies, and each category shares similar molecular substructures. We partition the pathway dataset category-wise as 80%/10%/10% into train/valid/test sets as partitions and follow the same training settings as the Pistachio.

## 4.3 Results

The performance of all methods is listed in Table. 2. For both the Pistachio and WuxiTest datasets, our approach achieves the highest success rate compared with the baselines. We observe that HgSearch achieves the best performance on the average cost metric in Pistachio, mainly from the near-exhaustive search performed on less challenging molecules. Our approach outperforms other baselines in average expansion by a large margin, demonstrating the performance gain in planning efficiency brought by RL training. In the WuxiTest dataset, our approach outperforms all other baselines in all four metrics. Since the WuxiTest dataset is designed to emphasize retrosynthesis strategies with more challenging but strategically similar molecules, the result proves that RL training can generalize planning knowledge from a molecule with similar substructures. We demonstrate the influence of time limit on the success rate for different approaches for the WuxiTest dataset in Fig. 3a for $N_{max} = 400$. We also demonstrate that the success rate tends to saturate when for $N > 400$ by extending to $N_{max} = 1000$ in Appendix C for all approaches.

---

[1] http://downloads.emolecules.com/free/2019-11-01/

| WUXITEST | RETRO* | HGSEARCH | DFPN-E | MCTS | GRASP GENERAL | GRASP EXPERT |
|---|---|---|---|---|---|---|
| SOURCE AVG. LENGTH | 7.50 | 7.65 | 7.29 | 8.15 | 7.05 | N/A |
| GRASP AVG. LENGTH | **7.35** | **7.55** | **7.07** | **7.55** | 7.05 | 7.20 |
| SOURCE AVG. RATING (0-10) | 7.6 | 8.1 | 7.4 | 6.5 | 8.3 | N/A |
| GRASP AVG. RATING (0-10) | **7.7** | 8.1 | **7.6** | **7.5** | 8.3 | **9.2** |

Table 3: Goal-driven planning performance summary. The experiment is conducted through a double-blind test with two different chemists to evaluate the quality of the pathway in terms of feasibility, efficiency, and simplicity.

**Goal-driven planning performance** Since GRASP is the first and only approach that empowers goal-driven planning, to evaluate whether GRASP is capable of generating high-quality goal-driven results, we conduct a double-blind user study of goal-driven planning on the WuxiTest. Specifically, we run GRASP using the building block molecules in the source pathways from different baselines as the GRASP's goal input to obtain a goal-driven result. In addition, we also include goal-driven planning using the general (GRASP's general planning without specifying goal) and expert (goal in the reference routes from the chemists) source pathways. The results in Table. 3 demonstrate that our approach can perform goal-oriented search and in the meantime generate a high-quality result. We provide an exemplar of pathway comparison in reference for demonstration in Fig. 5 and Fig. 6 in Appendix C.

**Compatibility to self-improved retrosynthetic planning** Self-improved retrosynthetic planning [10] is an end-to-end framework that fine-tunes the single-step model to imitate successful trajectories found by a fixed search (Retro* was used in the original work) algorithm by altering the prior distribution of single-step candidates in the search space. To evaluate the adaptation of GRASP with the self-improved framework, we follow the training procedure in [10] by replacing Retro* with GRASP and observe the performance on Pistachio and WuxiTest. As shown in Fig.3b, the self-improved retrosynthetic planning framework can improve the success rate of GRASP by using its own planning experience to fine-tune the single-step model.

## 4.4 Ablation studies

In this section, we investigate the following questions from different ablation studies: 1. The influence of two components: goal-driven relabeling (GDR) and self-imitation learning (SIL) over episodic reward during RL training. 2. How does the different probability of GDR affect the performance of general retrosynthesis planning and goal-driven retrosynthesis planning?

**Influence of different components:** We cross-check the training statistics of four combinations: GRASP without GDR and SIL, GRASP with GDR, GRASP with SIL, and GRASP with GDR and SIL. We evaluate the results on Pistachio by calculating the average reward with respect to training episodes. In binary reward setting, we use success rate as the criteria for reward evaluation, and the result is shown in Fig.3c. On the one hand, SIL significantly improves the overall training statistics but induces a more significant variance in the training process. The phenomenon is attributed to higher variance when using Monte-Carlo return and inevitable trade-offs in gradient propagation from different successful pathways under the same target. On the other hand, GDR also offers a certain amount of performance enhancement by relieving training difficulties induced by the sparse rewards. However, we are more interested in GDR's contribution to goal-driven planning.

**Influence of GDR probability:** The main hyperparameter we are interested in is GDR probability, which adjusts the distribution of transition tuples in the replay buffer for general and goal-driven planning. Specifically, we use the WuxiTest to evaluate the trade-off between general and goal-driven planning success rate for the hyperparameter controlling GDR probability. For goal-driven planning, we use the building block molecules in expert pathways for goal-driven input. The result is shown in Fig.3d. As expected, we observe that the success rate of goal-driven planning is lower than general planning, as it requires both general success and the specific goal reached. However, the success rate for goal-driven planning improves significantly when relabeling probability ranges from 10% to 70%. Nevertheless, high relabeling probability impairs the success rate for general planning since GDR might incur failures in general planning pathways, and increasing the proportion of goal-driven data leads to less proportion of general data. In conclusion, it is crucial to select an appropriate GDR probability depending on the actual usage of GRASP .

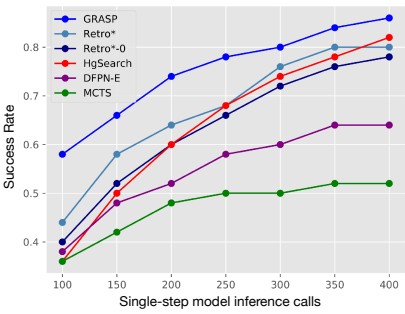

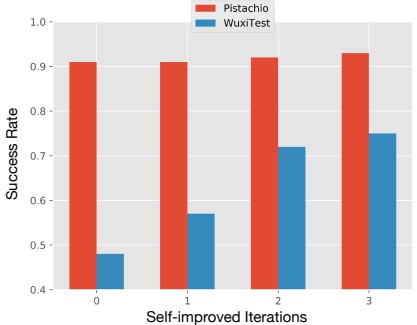

(a) The success rate under different limits of single-step inference calls

(b) The success rate with maximum 100 expansions vs. self-improved training iteration

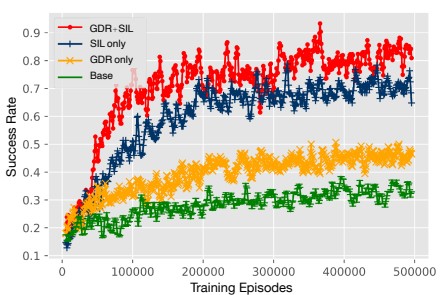

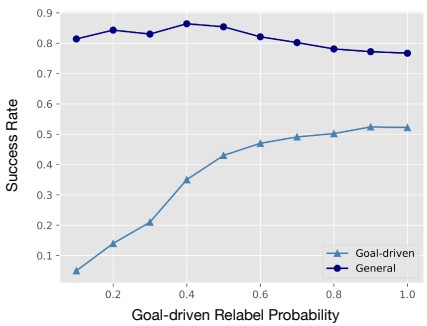

(c) The success rate vs. episodes during GRASP training

(d) The success rates vs. goal-driven relabeling probability

Figure 3: Experiments and ablation studies

# 5 Conclusion

This paper proposes GRASP, a novel goal-driven retrosynthetic planning approach. Unlike existing approaches that limit their generalization planning knowledge in a static dataset, GRASP can capture synthetic knowledge through self-generated experiences. Moreover, GRASP can perform goal-driven retrosynthetic planning that none of the existing approaches could explicitly accomplish. Experimental results on academic and industrial benchmark datasets demonstrate GRASP outperforms all baselines in general retrosynthetic planning and first achieves high-quality goal-driven planning. We have deployed GRASP in a real-world company to expedite the planning of synthetic pathways toward novel chemical compounds. By transforming the originally expertise-intensive pathway discovery process into efficient automation, GRASP significantly reduces the cost of the workforce on pathway design in both medicinal and process chemistry. More importantly, GRASP supports the high customization of goals and further alleviates the cost of human labor on pathway screening and post-processing. We believe that our work will significantly inspire related research on more efficient and chemists-machine interactive retrosynthetic planning frameworks. Nevertheless, the planning results from GRASP still need to be monitored or directed by personnel with chemical expertise if the target molecule is a novel or a rarely reported compound. The community lacks thorough research on the robustness of different single-step/multi-step retrosynthetic planning algorithms and how to generalize to out-of-distribution molecules, which we think could be an appealing research direction.

# 6 Acknowledgement

This work is sponsored by the Starry Night Science Fund at Shanghai Institute for Advanced Study (Zhejiang University) and Program of Zhejiang Province Science and Technology (2022C01044).

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
