# OpenReview forum: "GRASP: Navigating Retrosynthetic Planning with Goal-driven Policy"
_NeurIPS.cc/2022/Conference — NeurIPS 2022 Accept_

### Official Review · Reviewer_bEMH · 2022-07-11

**Rating:** 6
**Confidence:** 1
**Soundness:** 3 good
**Presentation:** 3 good
**Contribution:** 3 good

**Summary:**

This paper aims to solve retrosynthetic planning with RL. The authors proposed an actor-critic framework called GRASP, short for Goal-dRiven Actor-critic retroSynthetic Planning. This is claimed to be the first approach that enables goal-conditioned retrosynthetic planning, which mitigates the challenge in quality evaluation of pathways by directly fulfilling the requirements prescribed by chemists. The proposed method also leverage hindsight goal relabeling to speed up learning. They evaluate the performance of GRASP on both an academic and an industrial benchmark dataset (Pistachio and WuxiTest), and show superior results over all baselines.

**Questions:**

Nitpick:
* Line 189: "Segler et al." is missing in front of [24].

**Limitations:**

No discussion on limitations and potential negative societal impact at all.  Please add relevant pieces to the conclusion section.

**Strengths And Weaknesses:**

Disclaimer: the reviewer has zero background knowledge in retrosynthetic planning and thus is not able check whether the problem setting and evaluation are sensible or not, and whether the claim that "MCTS for retrosynthetic planning induces high-variance" is really a problem that cannot be easily mitigated. To my knowledge, modern approaches that use MCTS with limited rollout length and bootstrap with value network in general don't suffer much from this problem, but I don't know if there is anything specific to retrosynthesis as discussed in Sec. 3.3. I'm inclined to think this claim needs evidence but not sure whether that's obvious enough in this specific setting.

Otherwise, the proposed approach seems sound and mostly outperforms all baselines considered in this work (MCTS, DFPN-E, Retro*, Retro*-0, Hyper-Graph Search) on Pistachio and WuxiTest. The empirical evaluation seems comprehensive and appears to verify most of the main claims:
* Goal-driven planning mitigates the challenge in quality evaluation of pathways by directly fulfilling the requirements prescribed by chemists
* GRASP outperforms all baselines in general retrosynthetic planning metrics

---

> ### Author Response · Authors · 2022-08-02
> **Response to Reviewer bEMH**
>
> We sincerely appreciate your comments on this paper. You may find our response below for your major concern on the claim that "MCTS for retrosynthetic planning induces high-variance". We would appreciate it if you could let us know if you have any further concerns.
> ##### Q1: Whether the claim that "MCTS for retrosynthetic planning induces high-variance" is a problem that cannot be easily mitigated.
> > - There are two major reasons why MCTS for retrosynthetic planning (first proposed by Segler et al. [1\]) induces high variance.
> >
> >   - In [1], Segler et al. adopted an expensive **full rollout from vanilla MCTS** with a separate policy network rather than a bootstrapping value network for value estimation.
> >
> >   - In [1], each node (state) of the Monte Carlo tree consists of **a set of unsolved molecules** instead of a single molecule, which brings in a rollout space in the exponential rates. On the other hand, given the huge cost of online rollout, Segler et al. traded the quality of rollout estimation for the overall efficiency by rolling out only top-10 single-step candidates for each molecule and setting the maximum number of rollout iterations to be 50 only.
> >
> >   - Thus, the "high variance" problem is originated from both the variance induced in vanilla Monte-Carlo rollouts and the variation of the cardinality of the state representation.
> >
> > - This claim has also been substantiated in the Abstract and Introduction sections of Retro* \[2\] and serves as the major motivation behind the follow-up works of Segler et al. [1] to adopt a search tree with single molecule state representation and the bootstrapping value network for value estimation.
> >
> > - Empirical evidence: we have run the MCTS rollout for a randomly selected node (SMILES: C[C@H](c1ccccc1)N1C[C@]2(C(=O)OC(C)(C)C)C=CC[C@@H]2C1=S)	from USPTO. The average value variance for the expanded nodes is 0.152, which is pretty high compared with the value mean of 0.085.
> >
> > [1] Segler, Marwin, et al. "Planning chemical syntheses with deep neural networks and symbolic AI" Nature, 2018.
> > [2] Chen, Binghong, et al. "Retro*: Learning Retrosynthetic Planning with Neural Guided A* Search" ICML, 2020
> ##### Q2: Broader impacts and limitations
> > We have followed the reviewer's suggestion and included the potential negative societal impacts and limitations in the Conclusions section in the revised manuscript.

---

### Official Review · Reviewer_F5hv · 2022-07-12

**Rating:** 7
**Confidence:** 4
**Soundness:** 4 excellent
**Presentation:** 4 excellent
**Contribution:** 4 excellent

**Summary:**

The paper proposes a novel multi-step retrosynthetic planning framework called GRASP. The superiority of this reinforcement learning-based framework mainly relies on the capability of capturing synthetic knowledge through self-generated experiences. The paper is a pioneer work performing goal-driven retrosynthetic planning. In practice, the experiments demonstrate the effectiveness and the superiority of GRASP on both pathway datasets and expert datasets. The numerical results show GRASP achieves state-of-the-art performances on multi-step retrosynthetic planning tasks.

**Questions:**

Lack of ablation studies on different choices of single-step predictor. If it is possible to improve the performance by using a superior single-step predictor like Graph-to-Graph[1], or RetroXpert[2]?

[1] A Graph to Graphs Framework for Retrosynthesis Prediction, ICML 2020
[2] Decompose Retrosynthesis Prediction Like A Chemist, NuerIPS 2020

**Limitations:**

The broader impacts of this paper should be discussed.

**Strengths And Weaknesses:**

Strengths

+ The paper is well-written and well-organized. The notations and figures are easy to follow.
+ The idea is novel and works very well. It is a pioneer work performing goal-driven retrosynthetic planning.
+ The practical results demonstrate that the method could be helpful for chemists.
+ The analysis on the enhancement of GDR is convincing.
+ Codes are provided.

Weaknesses

- The performance depends on an upstream single-step retrosynthesis predictor. It would be better if this predictor could be optimized jointly with the RL-based framework.
- It seems like the expert datasets WuxiTest are not open-source resources.

---

> ### Author Response · Authors · 2022-08-02
> **Response to Reviewer F5hv**
>
> We thank the reviewer for the positive feedback and for highlighting our work's advantages. We address your concerns below point by point. Please kindly let us know whether you have any further concerns.
> ##### Q1: The performance depends on an upstream single-step retrosynthesis predictor. It would be better if this predictor could be optimized jointly with the RL-based framework.
> > We would like to share our insights as follows.
> >
> > - We have conducted the experiment in **Sec. 4.3**  "Compatibility to self-improved retrosynthetic planning" and **Fig. 3(b)** to **demonstrate that alternating optimization of the single-step predictor and GRASP indeed further improves the success rates**.
> >
> >   - Following [1], the training process alternates between (1) first training the GRASP for 1 full training cycle, and (2) then fine-tuning the single-step predictor with all single-step reactions in those **successful routes planned by GRASP**, which is equivalent to optimizing the single-step predictor with policy-gradient based methods (e.g., REINFORCE) in our RL-based framework. We have conducted 4 alternating iterations as such.
> >
> >   - Our previous attempt on **joint optimization of the single-step predictor during GRASP training of each route** brings even negative performance improvement, which is **attributed to (1)** the sub-optimal routes searched by GRASP may mislead the single-step predictor when GRASP does not converge, and **(2)** incremental training of the single-step predictor with online reactions poses catastrophic forgetting challenges.
> >
> >   - We are planning on designing a more efficient way to alternatively train the single-step and multi-step models from the continual learning perspective to improve our framework further.
> >
> > [1] Kim, Junsu, et al. "Self-improved retrosynthetic planning." ICML, 2021
> ##### Q2: Expert datasets WuxiTest are not open-source resources.
> > - Unfortunately, we are unable to open-source WuxiTest at this moment due to the strict regulation on IP. However, we have made steady progress on obtaining the IP approval from related parties, and will likely release the goal-driven dataset very soon.
> >
> > - Our algorithm has also been evaluated on the Pistachio and USPTO dataset both of which are open-source resources.
> ##### Q3: Ablation studies on different choices of the single-step predictor.
> > - As answered in Q1, our experiment on "Compatibility to self-improved retrosynthetic planning" in **Section 4.3 has demonstrated that improving the single-step predictor** via fine-tuning with the reactions in successful planning experiences is **indeed beneficial for improving planning efficiency**.
> >
> > - We have **implemented the current state-of-the-art single-step predictor, i.e., Dual-TF [2]**, to replace the template-based MLP single-step predictor on the USPTO dataset to incorporate our GRASP and Retro*. Moreover, we use the single-step "energy score" in Dual-TF to replace the single-step confidence score in GRASP and the single-step likelihood in Retro*. The results are shown in the following table, where we conclude:
> >
> >   - A better single-step predictor like Dual-TF as expected improves retrosynthesis planning, compared with the original MLP version single-step predictor (GRASP + MLP).
> >
> >   - Our multi-step planning framework GRASP is compatible with different single-step predictors, still outperforming Retro* with the SOTA single-step predictor Dual-TF.
> >
> >   - We would also like to share our insights on the reason why GRASP + Dual-TF does not improve significantly. The devil lies in **the gap between the single-step evaluation metric and the performance of single-step retrosynthesis in multi-step retrosynthesis planning**: the Top-k accuracy used for evaluating single-step predictors is calculated with matched reactions with the ground-truth in the single-step dataset, while most of the reactions in the optimal multi-step routes from a multi-step planning algorithm lie out of the distribution of the single-step dataset.
> >
> >   | Alg.             | Route Length | Succ Rate (Max iter. 100) |
> >   | ---------------- | ------------ | ------------------------- |
> >   | GRASP + MLP      |     6.17     |          90.52            |
> >   | GRASP + Dual-TF  |     5.72     |          91.58            |
> >   | Retro* + Dual-TF |     8.31     |          57.39            |
> >
> > [2] Sun, Ruoxi, et al. "Towards understanding retrosynthesis by energy-based models." NIPS 2021
> ##### Q4: Broader impacts and limitations
> > We have followed the reviewer's suggestion and included the potential negative societal impacts and limitations in the Conclusions section in the revised manuscript.

---

> > ### Comment · Reviewer_F5hv · 2022-08-07
> > **Responses to Authors**
> >
> > Thanks for the authors' well-written rebuttal. The authors almost address all my questions and concerns. However, I still haven't seen any words on the broader impacts or something like that in the revision?

---

> > > ### Author Response · Authors · 2022-08-07
> > > **Response to Reviewer F5hv**
> > >
> > > Thank you for letting us know your remaining concern on broader impacts. In the revised manuscript, we have further elaborated on the broader impacts of GRASP in Line 345-351.

---

### Official Review · Reviewer_5v4d · 2022-07-18

**Rating:** 6
**Confidence:** 3
**Soundness:** 3 good
**Presentation:** 3 good
**Contribution:** 2 fair

**Summary:**

This paper formulates retrosynthesis planning as a reinforcement learning problem and proposes a Goal-dRiven Actor-critic retroSynthetic Planning (GRASP) framework. They propose first to learn a policy network that takes continuous actions encoding the structure-level molecular information to allow navigation in the huge discrete action space of single-step reaction candidates. Then GRASP learns a goal-driven Q-value estimation network to update the policy, by sampling both successful and failed experiences and relabeling the goals of sampled experiences. The learned Q-value estimation and policy networks join to guide the Monte-Carlo Tree Search, after which GRASP returns diverse pathways due to a good exploration-exploitation tradeoff. They evaluated the performance of GRASP on both an academic and an industrial benchmark dataset. The results and user studies demonstrate that GRASP outperforms all baselines in general retrosynthetic planning metrics by a significant margin and is the first to achieve high-quality goal-driven retrosynthetic planning.

**Questions:**

Could the authors include more recent related methods and properly compare with them?

**Strengths And Weaknesses:**

The authors successfully adjusted RL to the retrosynthesis plan using the actor-critic RL method to learn from positive and negative experiences and navigate through the single-step reaction space.

They also applied goal-driven relabeling in retrosynthesis by exploiting the data generated from the retrosynthesis policy. This help to overcome the challenge in the quality evaluation of pathways.

The experiments are conducted on multiple datasets, including the Pistachio reaction dataset and a real-world expert dataset ‘WuxiTest’.
Ablation studies are included to investigate the effectiveness of different components.

For experimental comparison, the authors did not include recent related methods [1] as baselines, and still regard Retro* as the state of the art, which makes the significance of the proposed method less.

[1] Han, Peng, et al. "GNN-Retro: Retrosynthetic Planning with Graph Neural Networks." AAAI 2022.
[2] Xie, Shufang, et al. "RetroGraph: Retrosynthetic Planning with Graph Search." KDD 2022.

---

> ### Author Response · Authors · 2022-08-02
> **Response to Reviewer 5v4d**
>
> We sincerely appreciate your constructive comments on this paper. We detail our response below point by point. Please let us know if our response addresses the issues raised in this paper.
> ##### Q1: comparison with more recent related methods [1,2]
> > - We sincerely thank the reviewer for bringing our attention to these two valuable related works GNN-Retro [1] and RetroGraph[2] published after the NIPS submission deadline. We have updated our related work to include both works in Section 2 (Multi-step Retrosynthetic planning) in the revised manuscript.
> > - First of all, we would highlight the **unique contribution of GRASP in goal-driven retrosynthesis planning (named as GRASP Expert)**, which has not been considered and achieved before by previous works and [1,2].  However, goal-driven retrosynthesis (GDR) is essential for deploying retrosynthesis in a real-world pharmaceutical company, where chemists can customize their goals for affordable building block materials or easy-to-synthesize intermediate molecules. More detailed justification for GDR and the performance of **GRASP Expert** can be found in Line 44-55, Table 3, and Appendix Figure 6.
> > - Second, we have evaluated the proposed **GRASP General** (not goal-driven) on the **USPTO benchmark dataset** that has been adopted in both [1] and [2], instead of implementing [1,2] and evaluating on our benchmark datasets due to time limit in the response period. For fair comparison, we also **(a)** adopt the **template-based MLP single-step model** as in [1,2] and Retro* and **(b)** replace the MT-based confidence score in GRASP with the **likelihood from the MLP single-step model**. We report the full table below, and conclude similar observations as in Table 2.
> >   - GRASP General achieves **the shortest** Route Length, resulting from GRASP RL training that aims to prioritize shorter success routes with higher rewards. Shorter routes are easier to synthesize and are usually preferred, indicating the effectiveness of searched routes.
> >
> >   - GRASP General achieves **the best** performance on Success Rate under the max iteration (Max Iter.) of 100/200, and comparable performance on Success Rate under the max iteration of 300/400/500 (the small performance gap between ours and RetroGraph [2] at the max iteration of 500 is attributed to only 1 testing molecule). **Higher success rates within a small number of iterations (i.e., 100/200) are preferred** for retrosynthesis, indicating **higher efficiency**.
> >
> >   - Besides, RetroGraph [2] adopts GNN to evaluate the value of a molecule, and relies on a specific route dataset pre-generated by an A* search algorithm on the training set according to Pg.5, Sec. 3.3: "We use an offline-training strategy to train the GNN policy network. More specifically, we first use A* search on the training dataset to find synthetic plans for all target molecules". These two **contributions of RetroGraph[2] are orthogonal to our RL retrosynthesis planning framework**, and we are very interested in also implementing our value network with GNNs and trying this higher-quality training dataset in the near future to boost our algorithm further.
> >
> > | Alg.                      | Succ Rate (Max Iter. 100) | Succ Rate (Max Iter. 200) | Succ Rate (Max Iter. 300) | Succ Rate (Max Iter. 400) | Succ Rate (Max Iter. 500) | # of Iteration ↓ | Route Length ↓ | Route Cost ↓ |
> > |:------------------------- |:------------------------- |:------------------------- |:------------------------- |:------------------------- |:------------------------- |:---------------- |:-------------- |:------------ |
> > | Retro*-0    | 36.84  | 59.47  | 68.95  | 74.74     | 79.47    | 210.49   | 11.21 | 19.4   |
> > | Retro*  | 52.11    | 66.32  | 76.84 | 81.05 | 86.84   | 166.72  | 9.71 | 15.33  |
> > | Retro*+-0    | 67.37    | 82.1   | 93.16 | 95.26    | 96.32   | 96.14  | 7.69   | **11.66**    |
> > | Retro*+     | 71.05    | 85.26        | 88.95  | 90   | 91.05   | 100.15 | 8.74   | 15.23   |
> > | RetroGraph [2]            | 88.42   | **97.89**   | **98.95**   | **99.47**    | **99.47(189/190)**  | **45.13**    | 6.33  | 12.92|
> > | GNN-Retro (Threshold) [1] | N/A  | N/A       | N/A  | N/A   | 91.05   | N/A   | N/A    | N/A  |
> > | GNN-Retro (Embedding) [1] | N/A       | N/A      | N/A      | N/A   | 87.37    | N/A     | N/A    | N/A   |
> > | GRASP General       | **90.52**      | **97.89**       | 97.89        | 98.42      | 98.94(188/190)     | 48.47    | **6.17**   | 13.91   |
> >
> > [1] Han, Peng, et al. "GNN-Retro: Retrosynthetic Planning with Graph Neural Networks." AAAI 2022.
> > [2] Xie, Shufang, et al. "RetroGraph: Retrosynthetic Planning with Graph Search." KDD 2022.

---

### Meta-Review · Area_Chair_zhkD · 2022-08-25

**Recommendation:** Accept
**Confidence:** Less certain

**Metareview:**

This work proposes a goal driven actor-critic method for finding pathways with a specific prescribed goal such as building block materials. Various aspects like self-generation of data, hindsight goal relabeling while not novel by themselves, have found meaningful application on the chemical reaction tasks and the improved results has been appreciated by all reviewers. The authors are encouraged to incorporate reviewer feedback into account and also discuss the concurrent submissions as mentioned in the reviews for the sake of completeness in the camera-ready version.

**Award:**

No

---

### Decision · Program_Chairs · 2022-09-14

Accept